# In situ architecture of the algal nuclear pore complex

Shyamal Mosalaganti [1], Jan Kosinski[1,2], Sahradha Albert[3], Miroslava Schaffer[3], Daniela Strenkert[4], Patrice A. Salomé [4], Sabeeha S. Merchant [4], Jürgen M. Plitzko[3], Wolfgang Baumeister[3], Benjamin D. Engel [3] & Martin Beck [1,5]

Nuclear pore complexes (NPCs) span the nuclear envelope and mediate nucleocytoplasmic exchange. They are a hallmark of eukaryotes and deeply rooted in the evolutionary origin of cellular compartmentalization. NPCs have an elaborate architecture that has been well studied in vertebrates. Whether this architecture is unique or varies significantly in other eukaryotic kingdoms remains unknown, predominantly due to missing in situ structural data. Here, we report the architecture of the algal NPC from the early branching eukaryote *Chlamydomonas reinhardtii* and compare it to the human NPC. We find that the inner ring of the *Chlamydomonas* NPC has an unexpectedly large diameter, and the outer rings exhibit an asymmetric oligomeric state that has not been observed or predicted previously. Our study provides evidence that the NPC is subject to substantial structural variation between species. The divergent and conserved features of NPC architecture provide insights into the evolution of the nucleocytoplasmic transport machinery.

[1] Structural and Computational Biology Unit, European Molecular Biology Laboratory, Meyerhofstrasse 1, 69117 Heidelberg, Germany. [2] Hamburg Unit c/o DESY, European Molecular Biology Laboratory, Center for Structural Systems Biology (CSSB), Notkestrasse 85, 22607 Hamburg, Germany. [3] Department of Molecular Structural Biology, Max Planck Institute of Biochemistry, Am Klopferspitz 18, 82152 Martinsried, Germany. [4] Institute for Genomics and Proteomics, Department of Chemistry and Biochemistry, UCLA, 607 Charles E. Young Drive East, Los Angeles, CA 90095, USA. [5] Cell Biology and Biophysics Unit, European Molecular Biology Laboratory, Meyerhofstrasse 1, 69117 Heidelberg, Germany. These authors contributed equally: Shyamal Mosalaganti, Jan Kosinski, Sahradha Albert, Miroslava Schaffer. Correspondence and requests for materials should be addressed to W.B. (email: baumeist@biochem.mpg.de) or to B.D.E. (email: engelben@biochem.mpg.de) or to M.B. (email: martin.beck@embl.de)

Nuclear pore complexes (NPCs) mediate molecular traffic between the cytoplasm and nucleus, and are therefore indispensable for eukaryotic life. NPCs are built from ~30 nucleoporins (Nups) that are mostly conserved across eukaryotes, with some exceptions[1–3]. Nups are organized into various sub-complexes, which assemble together to form two outer rings that reside in the cytoplasm and nucleus, and an inner ring that fuses the inner and outer nuclear membranes. In the human NPC (HsNPC), the ten-protein Y-complex is a major component of the outer rings (also referred to as the cytoplasmic and nuclear rings). Thirty-two copies of the Y-complex arrange in a head-to-tail conformation to form concentric, reticulated rings within both the cytoplasmic and nuclear rings[4]. The Y-complex scaffold is complemented by additional subcomplexes that fulfill specific functions in the nuclear and cytoplasmic periphery and provide the directionality cue for nucleocytoplasmic exchange. The inner ring is composed of 32 protomers, each containing the Nup93 and Nup62 subcomplexes. Although the inner ring is constructed from different proteins than the outer rings, the oligomeric assembly of the inner and outer rings is similar[5,6].

Various biochemical and structural studies of NPC sub-complexes from vertebrates, fungi, and Trypanosomes have concluded that the subcomplexes are conserved (for a comprehensive review see ref. [7]). The Y-complex has been extensively analyzed in numerous species including the yeasts Saccharomyces cerevisiae[8] and Schizosaccharomyces pombe[9], the thermophilic fungi Chaetomium thermophilum[10] and Myceliophthora thermophila[11], and Homo sapiens[4]. Together, these studies concluded that the Y-shaped subcomplex architecture is conserved across distant branches of the eukaryotic tree of life. However, it remains unclear whether subcomplexes from different species assemble into NPCs in an identical fashion. This is highlighted by a prominent model proposed for yeast NPC architecture that suggests that yeast have fewer Y-complexes than humans[3]. Thus, the number of Y-complexes and the oligomeric state of the NPC across eukaryotic kingdoms remain uncertain. Cryo-electron tomography (cryo-ET) combined with subtomogram averaging[12] provides a powerful method to address this question by visualizing the in situ architecture of NPCs. Such in situ structural analysis has been performed for NPCs embedded within the isolated nuclear envelopes of HeLa cells[4,5,13] and Xenopus laevis[14], as well as for intact HeLa[15] and u2os cells[16]. Together, these studies showed that NPC architecture is consistent between vertebrates. Analyses of NPCs from the lower eukaryotes Dictyostelium discoideum[17] and S. cerevisiae[18] lacked the necessary resolution to visualize subcomplex architecture.

An important architectural feature underlying all previously proposed models of NPC architecture is the intrinsic C2 symmetry of the inner ring and Y-complexes across the plane of the nuclear envelope[1,3,7]. It has been proposed that the NPC's remarkable degree of symmetry might be essential to facilitate the modular assembly of its large macromolecular structure from a limited set of building blocks[19]. Here, we combine focused ion beam thinning of vitreous frozen cells[20–22] with in situ cryo-ET to analyze NPC architecture within the native cellular environment of Chlamydomonas reinhardtii, a unicellular green alga (Chlorophyte) and an early branching eukaryote. This approach facilitates structural analysis within intact cells in a close-to-living state without the need for subcellular fractionation or affinity purification. We find that the C. reinhardtii NPC (CrNPC) has several distinct architectural features, including an asymmetrical oligomeric state of the cytoplasmic and nuclear rings. We postulate that different mechanisms of Y-complex oligomerization might have evolved independently for the C. reinhardtii cytoplasmic and nuclear rings, and that NPC architecture may vary considerably throughout eukaryotic life.

## Results

### Key scaffolding subcomplexes are conserved in C. reinhardtii.

C. reinhardtii cells are particularly well suited for in situ structural biology, enabling high-resolution imaging of cellular structures[23–28]. This model organism is therefore an excellent candidate to address the question of how well current models of NPC architecture are transferable across eukaryotic species. We first explored the genome of C. reinhardtii[29] by sequence alignments to determine whether the key Nups of the NPC are detectable in the genome and whether the Nup subcomplexes are conserved. In agreement with a previous genomics study[30,31], we found homologs of all major scaffold and FG-Nups (Supplementary Fig. 1, Supplementary Table 1). The NUP188 gene, which was previously reported to be absent in plants[32,33], was present in the C. reinhardtii genome. We also detected a NUP188 homolog in the genome of Arabidopsis thaliana (At4g38760, in agreement with Neumann et al.[30,31] but in contrast to later studies[32,33]) emphasizing that Nup188 has a conserved role in the NPC scaffold architecture and is likely an ancient protein. Although sequence similarity cannot prove that one gene indeed encodes a functional equivalent of another gene, it is fair to conclude that the inner ring and Y-complexes are generally conserved in C. reinhardtii because the vast majority of their components that have been functionally analyzed in various species[7] were confidently detected.

However, we did not detect NUP358 and NUP153 genes, which in metazoa constitute cytoplasmic ring and nuclear ring-specific elements, respectively. The Y-complex member, NUP37, and the transmembrane Nups, GP210 and POM121, are also absent from the genome, whereas the chromatin-binding Nup, Elys, is encoded in a truncated form. Failure to detect these genes might be due to incorrect gene predictions in the current version of the genome. However, in the case of Nup358, it has been well established that this protein has evolved in animals and is absent in fungi and plants[34].

Based on the sequence alignments between the predicted C. reinhardtii, yeast, and human Nups, several intra- and inter-subcomplex contacts are conserved between these species. For example, patches of conservation between C. reinhardtii and human sequences can be found at several intra-Y-complex interfaces, including Nup160–Nup85 and Nup133–Nup107 (Supplementary Fig. 2a). Among the inter-subcomplex contacts, the IM-1 and IM-2 motifs[35,36] that link Nup93 to the Nup205 and Nup62 subcomplexes are conserved in CrNup93 (Supplementary Fig. 2b). Thus, the CrNPC inner ring may be arranged similarly to yeast and human inner rings. In addition, the interface between Nup107 and Sec13, responsible for the interaction between outer and inner copies of the Y-complex, is conserved in C. reinhardtii (Supplementary Fig. 2a), suggesting that the arrangement of double (inner and outer) Y-complexes occurs in the CrNPC.

To experimentally assess a functional relationship between the identified Nup-encoding genes, we analyzed whether they are co-expressed. We found that Nup-encoding genes are co-expressed during the synchronized diurnal cycle and peak at the end of the day (Supplementary Fig. 3). A large-scale expression analysis across various biological conditions revealed a Nup co-expression pattern that is highly distinct compared to the most closely related clathrin, COPI and COPII membrane-coating modules (Supplementary Fig. 4). This analysis indicates that the identified CrNups are genuine Nups and components of the same complex rather than of other vesicle systems.

### The algal NPC has a distinct architecture.

To analyze the in situ NPC architecture of C. reinhardtii, we acquired tomograms of the

nuclear envelope within its native cellular environment (Supplementary Fig. 5b) and extracted 78 subtomograms containing individual CrNPCs. We used subtomogram averaging to produce structural maps of the cytoplasmic, inner, and nuclear rings at an overall resolution of ~3 nm (Supplementary Fig. 5a, c)[28].

Comparison of the CrNPC to the HsNPC revealed striking differences in their overall dimensions and architecture (Fig. 1a). In humans, the outer rings are oriented in an upright position and are spatially separated from the inner ring by a connector element (green arrowheads, Fig. 1a)[13]. In C. reinhardtii, however, the outer rings are flatter and are directly stacked onto the inner ring. This direct engagement of inner and outer rings enforces a compact conformation of the CrNPC; the CrNPC scaffold extends only ~60 nm along the nucleocytoplasmic axis, whereas the HsNPC spans ~80 nm. While the outer diameters of the HsNPC and the CrNPC along the plane of the nuclear envelope are similar, the inner diameter of the CrNPC central channel is approximately 21 nm wider than that of the HsNPC (Fig. 1b), suggesting a modified inner ring arrangement. Lastly, the CrNPC's cytoplasmic ring has considerably less density than its nuclear ring. Such extensive asymmetric density across the nuclear envelope plane is surprising and has not been previously

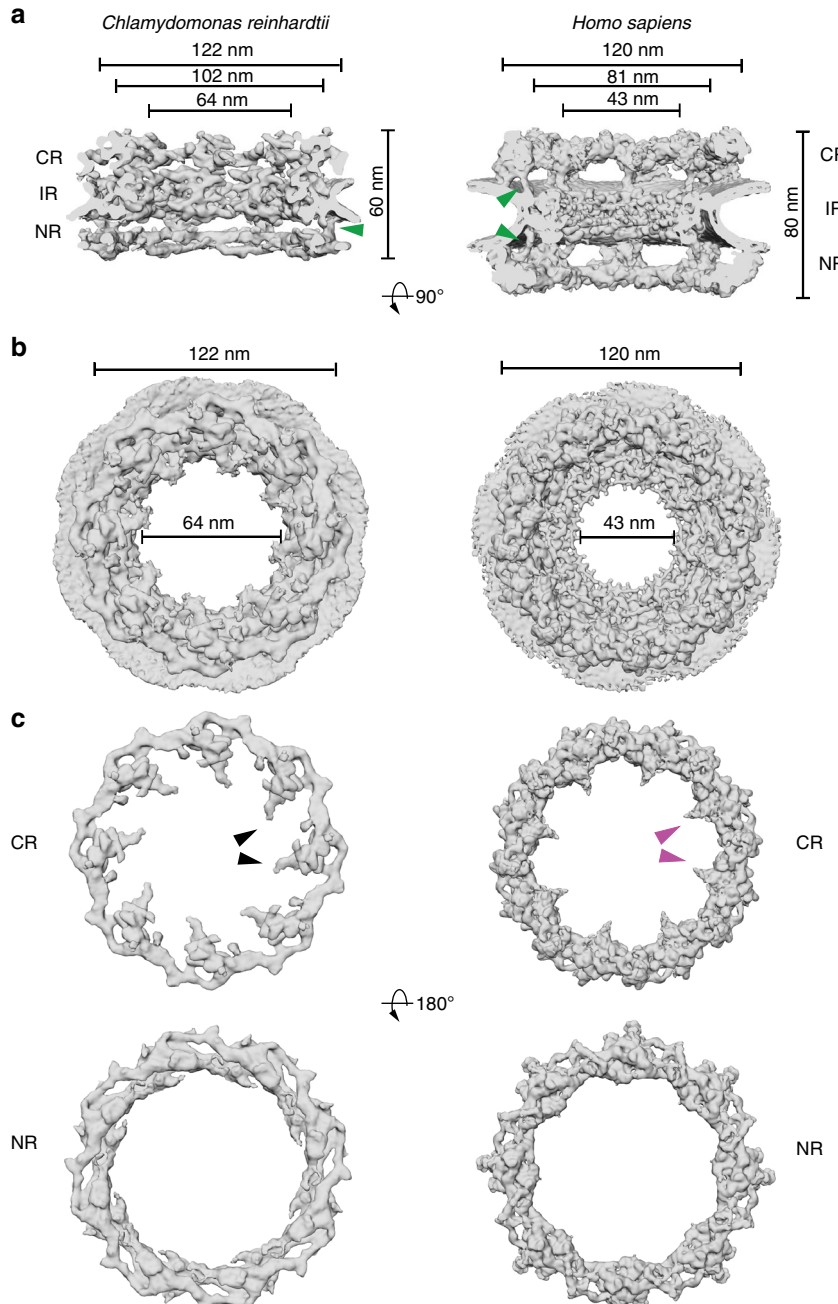

**Fig. 1** Structure of the CrNPC in comparison to the HsNPC. **a** Structures are displayed as rendered isosurfaces, sliced through the central axis. Green arrowheads indicate the connector element, which is absent from the cytoplasmic side of the CrNPC. **b** Cytoplasmic face view. The CrNPC central channel is dilated. **c** Cytoplasmic and nuclear rings of both NPCs. The CrNPC cytoplasmic ring has reduced density compared to the HsNPC. Black and magenta arrowheads indicate the density attributed to the Nup159 (Nup214 in humans) subcomplex, which forms cytoplasmic filaments that protrude towards the central channel. CR cytoplasmic ring, IR inner ring, and NR nuclear ring

reported for NPCs in any other organism (Fig. 1c). Although the cytoplasmic ring contains less density overall, it has distinct features within densities protruding towards the central channel (black arrowheads, Fig. 1c).

**The algal inner ring is dilated**. We next assessed whether the architectural arrangement of scaffolding Nup subcomplexes that we previously assigned into the *Hs*NPC[4,5] can explain the density observed for the subtomogram average of the *Cr*NPC. To this end, we applied a hierarchical procedure that included an unbiased fitting of low-pass filtered structural models of human Y-complexes and inner ring protomers. To evaluate the resulting fits, we used a recently proposed[37] scoring function that assesses both density cross-correlation and overlap with the tomographic map (Supplementary Note 1 and Supplementary Fig. 6). The resulting density assignment reveals that the *Cr*NPC map can be well explained by the structural repertoire of human scaffolding Nups (Supplementary Figs. 7–9), with some variations that are discussed below.

The 32 C2-symmetric protomers assigned to the *Hs*NPC inner ring[5,6] not only fit the inner ring of the *Cr*NPC, but also have an identical relative arrangement to that in the *Hs*NPC (Fig. 2a, Supplementary Fig. 7). The entire asymmetric unit, consisting of four C2-symmetric protomers, fits into the *Cr*NPC with high statistical significance. All four inner ring protomers fits were statistically significant after correction for multiple testing as assessed by systematic fitting (Supplementary Fig. 7). The density assigned to the inner ring protomers is weaker in the regions of the two inner protomers corresponding to the Nup62 subcomplex, leaving the exact number of Nup62 per asymmetric unit uncertain.

The arrangement of the four stacked protomers within each asymmetric unit of the inner ring (traditionally termed a spoke) is similar for the *Cr*NPC and *Hs*NPC (Fig. 2a). However, the eight spokes of the *Cr*NPC are positioned further apart, leading to an apparent dilation and wider central channel diameter (Fig. 2b). The tight interconnection between spokes observed in the *Hs*NPC is therefore relaxed in the *Cr*NPC, leading to gaps between the spokes that correspond to larger peripheral channels (black arrowheads, Fig. 2b). We conclude that although the principle composition and architecture of the inner ring within each asymmetric unit is conserved between these two distantly related eukaryotes, the overall spacing of the spokes is strikingly different.

**The algal cytoplasmic ring has a reduced oligomeric state**. We next examined the outer rings in detail. In humans, it has been

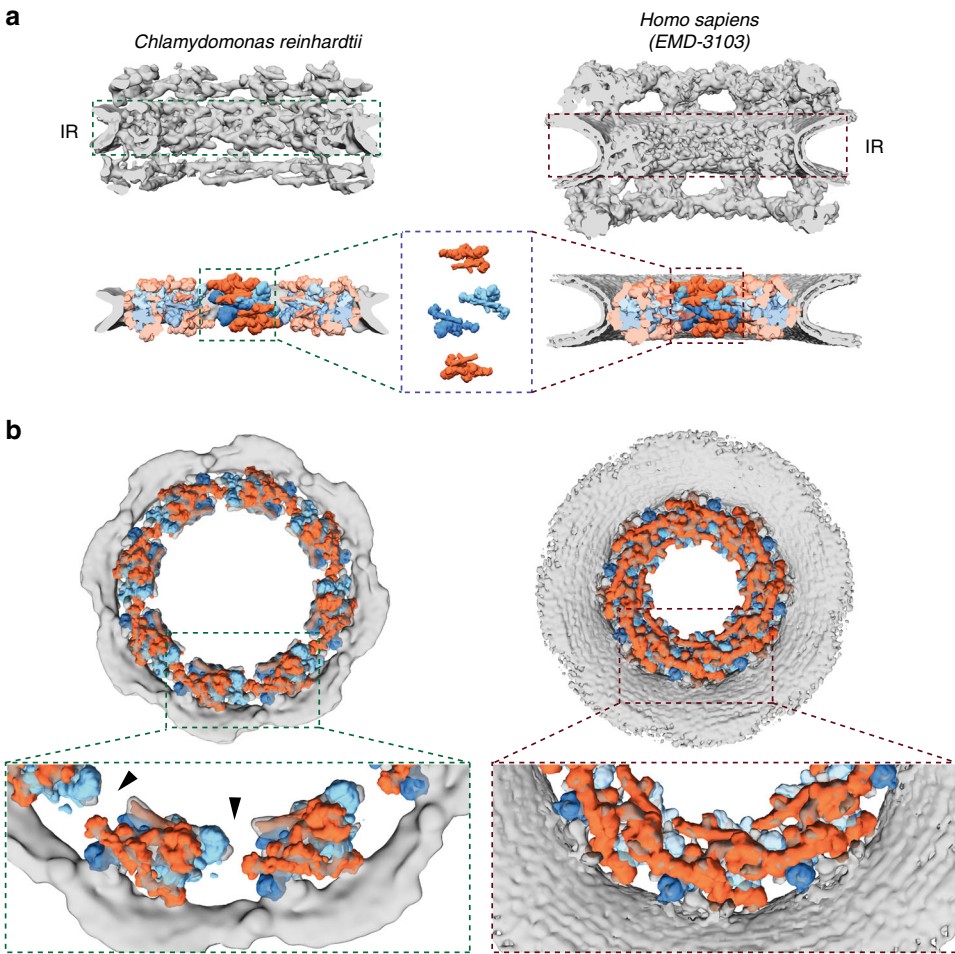

**Fig. 2** The inner ring of the *Cr*NPC is dilated compared to the *Hs*NPC. **a** Structures of the *Cr*NPC and the *Hs*NPC, displayed as rendered isosurfaces, sliced through the central axis. The inner rings (IR) are indicated with dashed boxes (top). The four protomers of the asymmetric unit (orange: outer protomers, blue: inner protomers), each containing Nup93 and Nup62 subcomplexes, explain the inner ring densities of both the *Cr*NPC (bottom left) and *Hs*NPC (bottom right). **b** View of the *Cr*NPC and *Hs*NPC inner rings seen along the nucleocytoplasmic axis. The asymmetric units (spokes) of the *Cr*NPC inner ring are separated from each other (left), leaving relatively large peripheral channels (arrowheads) between the spokes, whereas the spokes of the *Hs*NPC are positioned closer together (right)

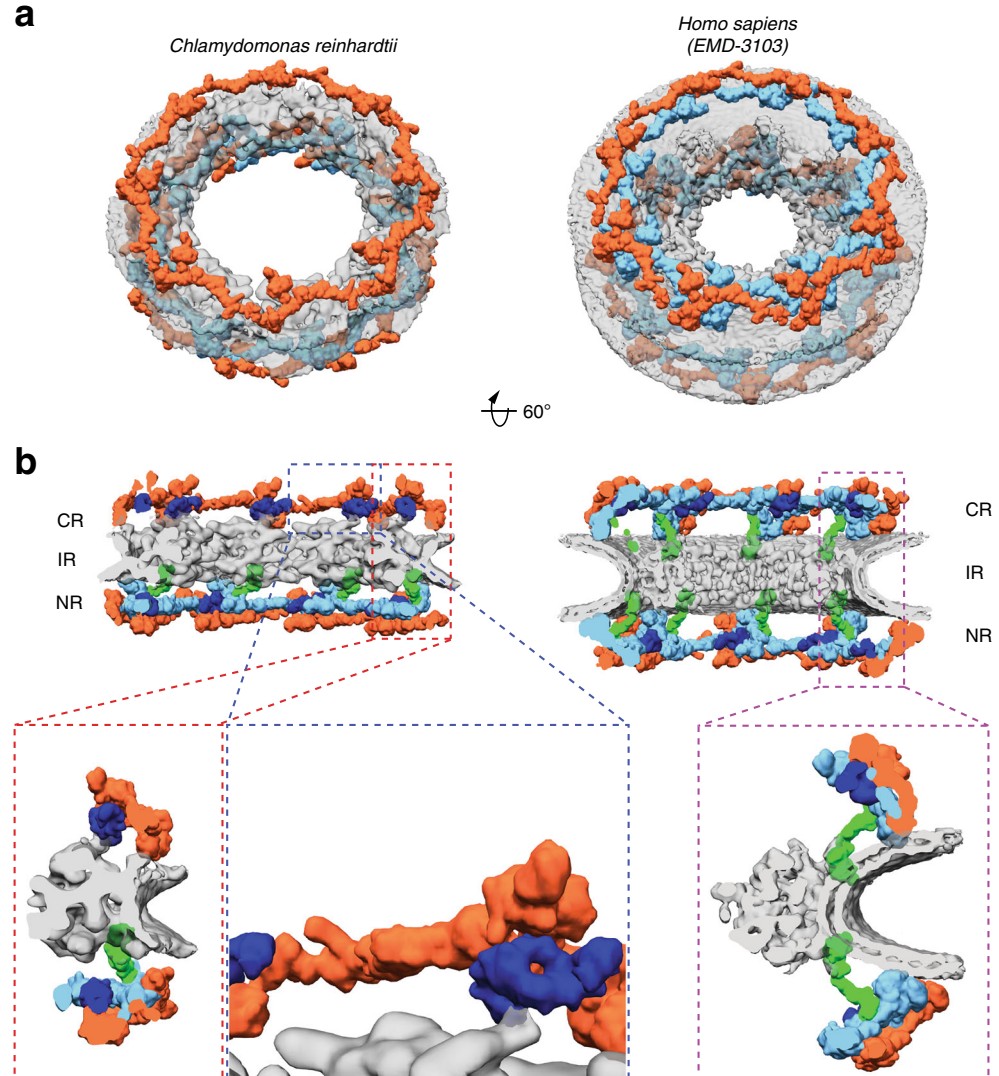

**Fig. 3** The *Cr*NPC has 24 Y-complexes. **a** Segmented Y-complexes according to the fits presented in Supplementary Figs. 6, 8, and 9 are shown superimposed with the inner ring structure (gray). The distribution of Y-complexes in the *Cr*NPC is asymmetric across the nuclear envelope plane. The cytoplasmic ring has only eight Y-complexes (orange), whereas the nuclear ring has 16 (orange and light blue). In the *Hs*NPC, the distribution is symmetric, with 16 Y-complexes in both of the outer rings. **b** Rotated views of the *Cr*NPC and *Hs*NPC, sliced through the central axis. Density attributed to large scaffold Nups (Nup205/Nup188) in the outer rings of the *Hs*NPC (dark blue) is found between the inner (light blue) and outer (orange) copies of the Y-complexes. Similar density is observed in the *Cr*NPC, although this assignment remains tentative at the given resolution. Enlarged views on the bottom row show the presence of only one connector element (green) in the *Cr*NPC (red box), with the cytoplasmic ring lacking the connector (blue box), while the *Hs*NPC contains two connector elements and duplicated Y-complexes (purple box). CR cytoplasmic ring, IR inner ring, and NR nuclear ring

established that the Y-complexes account for the majority of the observed outer ring density. In both the cytoplasmic and nuclear rings of the *Hs*NPC, 16 Y-complexes oligomerize in head-to-tail fashion to form reticulated double concentric rings[4] (Fig. 3a). We confidently identified Y-complexes at the expected positions in the *Cr*NPC map (Fig. 3a) using exhaustive fitting of low-pass-filtered human models (Supplementary Figs. 6, 8, and 9). We repeated this procedure with the structural model of the yeast Y-complex[36] and obtained highly similar results (Supplementary Figs. 8 and 9). The map of the *Cr*NPC is further consistent with various other features of Y-complex architecture, such as the overall length of the Y-complex, proximity of the Y-complex's large arm and tail to the membrane, and the head-to-tail contact of rotationally sequential Y-complexes (Supplementary Fig. 10). Although the identification of individual

Y-complexes within the *Cr*NPC is unambiguous, there are overall conformational differences compared to the *Hs*NPC. In both the cytoplasmic and nuclear rings of the *Cr*NPC, the Y-complexes are tilted down towards the dilated inner ring, resulting in flatter outer ring architecture than in the *Hs*NPC (Fig. 3b, Supplementary Movie 1).

We found that two key features of outer ring architecture are missing from the *Cr*NPC. First, the cytoplasmic ring contains only eight Y-complexes, half the number present in the *Hs*NPC (Fig. 3a). We confirmed this finding by (i) calculating a difference map between the nuclear and cytoplasmic rings, which leaves behind a cryo-EM density corresponding to eight Y-complexes (Supplementary Fig. 11), and (ii) exhaustive fitting analysis of the human double Y-complex into the *Cr*NPC, where the significant hits are only detected in the nuclear ring and not in the

cytoplasmic ring (Supplementary Figs. 8 and 9). This explains why the cytoplasmic ring has less density in comparison to the nuclear ring in the *Cr*NPC map (Fig. 1c). The *Cr*NPC nuclear ring contains 16 Y-complexes that are in rotational register with the 16 inner and outer Y-complexes of the *Hs*NPC nuclear ring, while the *Cr*NPC cytoplasmic ring contains eight Y-complexes that are in rotational register with the outer copies of the Y-complexes in the *Hs*NPC cytoplasmic ring (Supplementary Fig. 12). In total, *C. reinhardtii* has only 24 Y-complexes, which are asymmetrically distributed across the nuclear envelope plane, in contrast to any previously proposed model of NPC scaffold architecture (Fig. 3, Supplementary Movie 1). This oligomeric state is consistent with the finding that metazoan-specific Nup358, which is required for linking the inner and outer Y-complexes of the cytoplasmic ring in humans[13], is absent in algae (Supplementary Fig. 13).

Second, the connector density attributed to *Hs*Nup155 in the *Hs*NPC[13] is missing from the cytoplasmic but not the nuclear side of the *Cr*NPC (Fig. 3b). This is surprising because the connector is the only rigid structural element that connects the inner ring to the outer rings in the *Hs*NPC. We therefore inspected the contact points between the inner and cytoplasmic rings of the *Cr*NPC. We found that contact is made by densities attributed to large scaffold Nups (Nup188 or Nup205) in the cytoplasmic ring of the *Hs*NPC (Fig. 3). We conclude that although the *C. reinhardtii* Y-complexes of the cytoplasmic ring arrange in a head-to-tail fashion similarly to humans, neither the oligomeric state nor the connection to the inner ring is conserved between algae and humans.

**Cytoplasmic subcomplexes differ between algae and humans.** The Nup214 subcomplex (Nup159 subcomplex in fungi) is a key player in the remodeling and export of messenger ribonucleo-protein particles[1]. It is a major component of the cytoplasmic filaments that decorate the NPC scaffold at the cytoplasmic ring. In both the *Cr*NPC and *Hs*NPC, we observed characteristic densities extending from the cytoplasmic ring towards the central channel. However, the two densities are considerably different. The density protruding from the *C. reinhardtii* cytoplasmic ring is relatively large (black arrowheads, Fig. 1c) and would be consistent with previous analysis based on subtomogram averaging and cross-linking mass spectrometry that has associated the Nup159 subcomplex with the small arm of the Y-complex[4,38,39]. The corresponding density protruding from the human cytoplasmic ring is smaller (magenta arrowheads, Fig. 1c). This may be due to flexibility or a different subcomplex oligomeric state, emphasizing species-specific differences of this rather poorly conserved NPC module. At the given resolution, neither the algal nor the *Hs*NPC density map enables fitting the dimeric yeast Nup159 subcomplex[38,39]. Taken together, this analysis suggests that not only the Y-complexes but also more peripheral subcomplexes are subject to extensive structural variation across the tree of life.

## Discussion

The evolution of the NPC is deeply rooted in the origin of eukaryotes. The protocoatamer hypothesis suggests that NPCs and trafficking vesicles arose from a common ancestor by divergent evolution[40]. Understanding the evolution of the NPC is therefore pivotal for addressing the origin of eukaryotic compartmentalization. Although most Nups are postulated to be ancient proteins[30], it remains unclear to what extent the organizational principles of the NPC are conserved in subsequent eukaryotic lineages. Here, by comparing NPCs of species from two distant eukaryotic kingdoms, we find that the oligomeric state of the NPC can vary substantially.

Our findings are derived from the *Cr*NPC structure obtained by in situ cryo-ET. Analysis of the *C. reinhardtii* genome reveals that this alga has orthologs of all known Nups required to form the major scaffold subcomplexes of the NPC. Transcriptomic analysis strongly suggests that these Nups form functional complexes together, as they are tightly co-expressed across various biological conditions. Our systematic fitting of the *Cr*NPC with yeast and human structures supports the conclusion that the *Cr*NPC is built from scaffold subcomplexes that are compositionally and structurally similar to human and yeast subcomplexes but assembled with a distinct stoichiometry. While we cannot exclude that the compositional variability of the *Cr*NPC extends even further (e.g., through unidentified Nups specific to algae), the assignment at the level of subcomplexes already reveals striking features.

In particular, the density map reveals that the *Cr*NPC contains a high degree of asymmetric density, with a total of only 24 Y-complexes, highlighting the importance of asymmetric linker Nups that are required to connect scaffold Nups[35]. Interestingly, a recent biochemical and morphological study of the Trypanosome NPC suggests that its NPC structure may be highly symmetric[7]. Although one might hypothesize that only 16 Y-complexes were present in the outer rings of ancient NPCs, with the same stoichiometry as proposed for the yeast NPC[41], it remains unclear whether the asymmetric oligomeric state of the *Cr*NPC arose due to a loss or a gain of function; i.e., it is not clear whether the *Cr*NPC evolved from NPCs with 16 or 32 copies of the Y-complex, or if 24 copies might even correspond to an ancient oligomeric state. Since a highly similar mode of nuclear Y-complex duplication is found in algae (*C. reinhardtii*) and vertebrates (humans), we consider it likely that vertebrates have duplicated their cytoplasmic Y-complexes using protein–protein interfaces that had already evolved for the nuclear ring, but using the metazoan-specific Nup358 as a dimerizer[13]. Such oligomeric duplication events are frequently observed during the evolution of protein complexes[42].

The inner ring of the *Cr*NPC map is dilated in comparison to the *Hs*NPC map, with substantial spacing between its rotationally symmetric spokes, thereby forming relatively large peripheral channels that have been proposed to accommodate the import of inner nuclear membrane proteins[43]. The *Cr*NPC inner ring has the same diameter as the outer rings, which are horizontally stacked upon it. In this conformation, the head-to-tail connection of the outer ring Y-complexes might be important for restricting the maximal dilation of the pore. Are these species-specific differences in the inner ring, or could they be related to the NPC's functional state? Independent cryo-ET structural analysis suggests that such elaborate conformational changes might also occur in vertebrates[12]. Constricted inner ring conformations have been observed not only in isolated *X. laevis* and HeLa cell nuclear envelopes but also within intact u2os cells[16], while more dilated conformations were observed within intact HeLa cells[15]. Taken together with our data from intact *C. reinhardtii* cells, these findings suggest that both constricted and dilated conformations have physiological relevance. We speculate that not only the FG-rich regions, but also the scaffold of the NPC may be much more dynamic than anticipated. Previous studies have reported the dilation of isolated *X. laevis* NPCs upon treatment with chemicals such as *trans*-cyclohexane-1,2-diol and steroids[44,45]. Using atomic force microscopy, these studies found that the NPC central channel diameter can expand up to 63 nm, the same diameter that we observed in *C. reinhardtii*. How such massive conformational changes are structurally induced and potentially regulated awaits further analysis. The local FG-Nup

concentration within the central channel might change during inner ring dilation. It remains to be determined whether inner ring dilation has any effect on nucleocytoplasmic transport activity, such as the rates and size limits of the transiting substrates, or whether it is relevant for inner nuclear membrane protein import.

Using in situ cryo-ET enabled by cryo-FIB milling, we were able to identify major structural variations within the NPC. Our study therefore underscores the importance of structural analysis within the native cellular environments of divergent species to understand the breadth of NPC architecture and ultimately gain insights into both NPC function and evolution.

## Methods

**Cryo-ET**. Cells were prepared for data acquisition based on procedures described in Schaffer et al.[46]. Briefly, cells were blotted onto EM grids, which were plunge-frozen into a liquid ethane/propane mixture using a Vitrobot mark IV (FEI) and then transferred onto a cryo stage in a Scios (FEI) or Quanta (FEI) FIB/SEM microscope. Cells were thinned with a gallium ion beam and transferred into a Titan Krios transmission electron microscope (FEI) equipped with a K2 Summit camera (Gatan) for tilt series acquisition (Supplementary Table 2). This data set has been also analyzed by Albert et al.[28].

**CrNPC structure determination**. Tomogram reconstruction and subtomogram averaging of the CrNPC is described in an parallel study[28]. Briefly, 78 NPCs were picked from twice-binned tomograms. Particles were manually aligned for correct orientation of the cytoplasmic and the nuclear rings of NPCs. An initial average of the whole NPC with imposed eight-fold symmetry was calculated using PyTom software[47]. The eight asymmetric units of the individually aligned NPCs were then extracted, yielding 624 asymmetric units. Alignment and averaging of these asymmetric units were carried out using the AV3/TOM packages[48], using iterative missing wedge weighted subtomogram averaging. After a few iterations of the entire asymmetric unit, masks specific to the cytoplasmic, nuclear, and inner rings of the asymmetric unit were used to further align each of these parts separately[4].

**Identification of C. reinhardtii nups**. The C. reinhardtii Nups were identified by retrieving predicted Nup sequences from the Phytozome platform[49] based on annotations or by BLAST[50] searches of human and plant Nups against the C. reinhardtii genomic sequence and predicted protein database. All identifications were confirmed by reverse BLAST searches (using the predicted Nups as queries), searches against a non-redundant protein database, and by domain mapping using the HHpred server[51] to ensure that the identified genes are bona fide Nup orthologs rather than more remote homologs from other families (e.g., vesicle coat proteins).

**Culture conditions for cell cycle experiments**. C. reinhardtii strain CC-5390 was grown synchronously[52] in HSM medium[53], in a pre-sterilized flat panel photo-bioreactor operated in turbidostat mode, aerated, and mixed with pressurized air at an airflow of 0.2 L min$^{-1}$. The temperature was maintained at 28 °C during the day (average irradiance of 200 μmol photons m$^{-2}$ s$^{-1}$) and at 18 °C during the night cycle.

**Co-expression analysis**. Sequencing reads for 518 samples from 60 independent experiments were downloaded from NCBI Short Read Archive and mapped to version v5.5 of the Chlamydomonas genome. These experiments can be broadly divided into 27 nutrient experiments (nitrogen deficiency and resupply, acetate resupply, micronutrient [Fe, Cu, Zn] deficiency, and resupply), eight development experiments (including diurnal cycles, cell wall generation, sexual cycle, and deflagellation), nine signaling experiments (response to bilin treatment, responses to high light, miRNA regulation, and photoperiodic signaling), six stress experiments (responses to $H_2O_2$, Cd, Ni, or Rose Bengal), and ten additional experiments conducted in selected mutant backgrounds or C. reinhardtii strains. Expression estimates for each gene were then subjected to (i) $log_2$ normalization (as FKPM [fragments per kilobase of transcripts per million mapped reads] +1 to account for genes with zero counts in some samples), (ii) quantile normalization to fit all samples to a single distribution density with the R package preprocessCore, and (iii) subtracting quantile-normalized gene mean across all samples. COPI, COPII, and clathrin genes were identified as above for Nups. Note that Sec16 could not be confidently identified and thus is not included in the analysis. Co-expression matrices for each protein complex (NPC, COPI, COPII, and clathrin) were plotted with the R package corrplot, and genes were ordered by hierarchical clustering (hclust, "complete" method). Only PCC values with an associated p-value ≤ 0.01 were plotted. The final correlation matrix was replotted with all genes at once, in the order provided by hclust. Distributions of PCCs for various parts of the correlation matrix were plotted in R with the density function.

**Assignment of subcomplexes within the CrNPC map**. To assign densities of the CrNPC map to specific subcomplexes, a hierarchical fitting procedure (Supplementary Fig. 6) was applied, which is described in detail in Supplementary Note 1.

**Data availability**. Cryo-EM maps of the C. reinhardtii nuclear pore complex and the corresponding cytoplasmic, inner, and nuclear rings have been deposited in the EMDB with the accession codes EMD-4355, EMD-4332, EMD-4333, and EMD-4334, respectively. All other data that support the findings of this study are available from the corresponding authors upon reasonable request.

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

## Acknowledgment

We thank Drs. Felix Willmund and Jacob Musser for advice. Christian Zimmerli, Marc Wehmer, Drs. Elizabeth Villa, Eri Sakata, and Matteo Allegretti are acknowledged for help with the preparation of this manuscript. S.M. and J.K. were supported by the EMBL Interdisciplinary Postdoc Programme under Marie Curie COFUND Actions. J.K. acknowledges the financial support by the BMBF in the framework of the project/FKZ 031L0100. W.B. acknowledges funding from the Max Planck Society and Deutsche Forschungsgemeinschaft excellence clusters CIPSM and SFB 1035. M.B. acknowledges funding by EMBL and the European Research Council (309271-NPCAtlas and 724349-ComplexAssembly). S.S.M., P.A.S., and D.S. were supported by a cooperative agreement with the US Department of Energy Office of Science, Office of Biological and Environmental Research program under Award DE-FC02-02ER63421.

## Author contributions

FIB milling: M.S.; cryo-ET: M.S., B.D.E., and S.A.; structural analysis: S.M. and S.A.; structural modeling: S.M. and J.K.; bioinformatic analysis: J.K.; co-expression analysis: D.S., P.A.S., and S.S.M.; project management: S.S.M., J.M.P., W.B., B.D.E., and M.B.; manuscript writing: S.M., J.K., W.B., B.D.E., and M.B.

## Additional information

**Competing interests:** The authors declare no competing interests.

