## [Peer Review File · Nature Communications]

Reviewers' comments:

Reviewer #1 (Remarks to the Author):

In this manuscript, Mosalaganti and colleagues characterize the structure of the nuclear pore complex (NPC) from the unicellular algae *Chlamydomonas reinhardtii*. Using ion beam thinning, NPCs were imaged in situ without the need of any biochemical purification. By subtomogram averaging the authors could produce NPC maps with a resolution of approximately 30Å. These maps reveal very intriguing differences in the overall organization of the *Chlamydomonas* NPC and the previously characterized human NPC including major differences in the overall arrangements of the various organizational units. Furthermore, the algal NPC is significantly dilated and the central opening is ~50 % wider in *Chlamydomonas* than in humans (64 nm vs 43 nm). These differences are remarkable and very interesting since individual NPC building blocks are overall well conserved from algae to yeast (Fig. 1). These results show that the NPC architecture highly diverged in the course of evolution.

The authors then go on to fit individual subcomplexes into the map and conclude that the inner ring has the same basic organizational principles as in humans whereas the outer rings are distinct with 16 copies of the Y complex on the nuclear side and 8 copies on the cytoplasmic side (in humans 16 copies were assigned on both sides). For this, low pass filtered models of human subcomplexes were initially fit into the map and then more or less manually adjusted to make the fits work. I find this procedure quite unsatisfying especially given the fact that in some cases the initial, unbiased fits were not statistically significant. Thus the key assumption that underlies this molecular model of the *Chlamydomonas* NPC is that the architectural organization of the key Nup subcomplexes is fully conserved between algae and humans. But can this assumption be made given the large differences in the overall architecture? I feel that additional follow up experiments would be needed to verify some of the key conclusions. For example, can the Y copy number (24 versus 32) be confirmed? Are the major biochemical interactions (that link individual subcomplexes within the NPC) conserved between algae/yeast/human or is the overall layout flexible (as the large divergence might suggest)?

Reviewer #2 (Remarks to the Author):

Mosalaganti et al., describe the in situ structure of the nuclear pore complex from the green algae *Chlamydomonas reinhardtii* (CrNPC). The structure was obtained by aligning 78 subvolumes of nuclear pore complexes and then applying 8-fold symmetry. The resulting average shows clear structural features to 30 Å resolution allowing direct comparison with the human NPC (HsNPC). The authors discovered that while the overall architecture of the human and green algae NPCs were similar, several differences occurred at the subunit level. These evolutionary differences will greatly assist in deciphering how nuclear pores function.

The manuscript is well written with clear figures and explanations. The cryo-ET and subtomogram averaging is of high standard, the fitting of the atomic structures is clearly explained and the conclusions justified. The data presented is of high interest to the people in the NPC field and also

researchers interested in the evolution of eukaryotes. Thus, I strongly recommend the publication of this manuscript.

Minor comments:

line 243: remove 'is'

Line 262: remove 'be'

Supplementary figure 1: Perhaps choose a different color for Elys (e.g. green) as it took a while to find.

Reviewers' comments:

Reviewer #1 (Remarks to the Author):

In this manuscript, Mosalaganti and colleagues characterize the structure of the nuclear pore complex (NPC) from the unicellular algae *Chlamydomonas reinhardtii*. Using ion beam thinning, NPCs were imaged in situ without the need of any biochemical purification. By subtomogram averaging the authors could produce NPC maps with a resolution of approximately 30Å. These maps reveal very intriguing differences in the overall organization of the *Chlamydomonas* NPC and the previously characterized human NPC including major differences in the overall arrangements of the various organizational units. Furthermore, the algal NPC is significantly dilated and the central opening is ~50 % wider in *Chlamydomonas* than in humans (64 nm vs 43 nm). These differences are remarkable and very interesting since individual NPC building blocks are overall well conserved from algae to yeast (Fig. 1). These results show that the NPC architecture highly diverged in the course of evolution.

We thank the reviewer for the enthusiasm towards our results.

The authors then go on to fit individual subcomplexes into the map and conclude that the inner ring has the same basic organizational principles as in humans whereas the outer rings are distinct with 16 copies of the Y complex on the nuclear side and 8 copies on the cytoplasmic side (in humans 16 copies were assigned on both sides). For this, low pass filtered models of human subcomplexes were initially fit into the map and then more or less manually adjusted to make the fits work. I find this procedure quite unsatisfying especially given the fact that in some cases the initial, unbiased fits were not statistically significant.

We have addressed this by implementing an improved fitting procedure that we explain below.

At the outset, we want to clarify that the procedure we had used in the original version of our manuscript was appropriate. Although some of the fits of Y complexes were not statistically significant during our exhaustive fitting procedure (based on our previously published protocol), they arise as the only possible solution after filtering for significant clashes between the fits and large overlaps with the membrane (Supplementary Fig. 3 in the previous version of the manuscript). Moreover, the adjustments of fits were performed only after performing the entire hierarchical fitting procedure, so 60-80% of the Y-complex always fit to the map without these adjustments. Also, the adjustments were limited to splitting the structures at the hinges between domains and local fitting of the domains. The local fitting always kept the fits close to the initial rigid body fit of the entire subcomplex. We apologize for the lack of clarity on this.

We wish to emphasize that one of the reasons for non-significant top fits is imperfect empirical null distribution used for calculating the p-values. If we filter the fits loosely (as in our previous publications), with a parameter that selects fits in which at least 30% of the fitted model is covered by electron density (defined at very permissive threshold), we obtain empirical null distribution that contains many fits with high normalized cross-correlation values but little overlap with the map, which consequentially reduces the p-values of the top fits. This can be accounted for by setting the fitting parameters more stringently, such that at least 50-70% of the fitted model must be covered by the density in all considered fits. In such a scenario, however, almost only true positive fits (i.e. fits corresponding to the arrangement known from human NPC) of the Y-complex are identified, and the total number of fits

becomes so small that it is impossible to estimate the empirical null distribution in order to determine statistical significance. Nonetheless, this does not change the fact that the highest scoring fits account for the three Y-complexes per asymmetric unit (one in the cytoplasmic ring and two in the nuclear ring).

Inspired by the reviewer's comment, we nevertheless looked for a more elegant solution for this problem. Maya Topf's laboratory recently proposed a scoring function that takes into account both cross-correlation and the overlap between fitted structure and the targeted map (Joseph et al. JSB 2017). We also realized that a considerable fraction of the false positive fits simply accounted for Y-complexes flatly aligned into the nuclear membranes that are also part of the Chlamydomonas map. We therefore implemented the new scoring function and applied it to a Chlamydomonas map in which the nuclear membranes were segmented out. This analysis yielded highly significant fits in two positions in the nuclear and one position in the cytoplasmic ring (per rotational segment) and strongly supports our conclusions. We repeated this analysis with the yeast Y-complex structure (Stuwe et al. Science 2015) and obtained highly similar results. We therefore conclude that our fitting analysis is robust (Supplementary Figs 8 and 9).

Furthermore, we included additional analysis to support the key conclusions of our manuscript. First, we show that the sequences, which were previously predicted to be important for inter-subunit contacts among the Y-complex are conserved in the Chlamydomonas (Supplementary Fig. 2). Second, a difference map of the cytoplasmic and nuclear rings clearly identifies one Y-complex and the density likely accounting for the Nup214 complex as the key differences between the two rings (Supplementary Fig. 11). Finally, We also fitted the double Y-complex arrangement (inner and outer copy) from our structural model of the human NPC into the Chlamydomonas map and obtained significant hits only in the nuclear but not in the cytoplasmic ring, as expected (Supplementary Figs. 8 and 9).

Finally, we want to stress that the observed electron density in the outer rings of the Chlamydomonas map is consistent with known properties of NPC architecture in various ways that are not considered by the fitting procedure:

- 1. The proximity of the Nup160 arm and Nup133 to the membrane, and to each other (head to tail contact of rotational consecutive Y-complexes) is clearly observed as highlighted in Supplementary Fig. 10.*
- 2. The overall length of the Y-complex is consistent with previous models.*
- 3. Additional density pointing inwards from the Nup85 arm (Bui et al. Cell 2013; Fernandez et al., Cell 2016) most likely corresponding to the Nup214 complex is clearly observed and exclusive to the cytoplasmic ring.*
- 4. The rotational registration of the inner Y-complexes to each other and to the inner ring complexes is consistent with previous models (Supplementary Fig. 12)*

We made an effort to explain these points better in the revised manuscript.

Thus the key assumption that underlies this molecular model of the Chlamydomonas NPC is that the architectural organization of the key Nup subcomplexes is fully conserved between

algae and humans. But can this assumption be made given the large differences in the overall architecture?

The purpose of the systematic fitting procedure was to address to what extent architectural features observed by our in situ structural analysis in humans are consistent with the map of the Chlamydomonas NPC. For the inner ring, the similarity is very high. For the outer rings, considerable differences are observed, as we discuss in our manuscript. However, structures of the human and yeast Y-complexes fit into the Chlamydomonas NPC map with statistical significance, demonstrating the conservation of Y-complex architecture. The Y-complex has been structurally and biochemically analysed in baker's yeast, fission yeast, filamentous fungi, frogs, and humans and its key structural features were always shown to be conserved. We have made an effort to stress this key point in the revised version of the manuscript, and have revised portions of the introduction and results accordingly.

In the revised manuscript, we have added gene-expression analysis to further validate our bioinformatics identification of conserved Chlamydomonas nucleoporin-encoding genes. Comparing these nucleoporin-encoding genes to the evolutionarily most closely related genes encoding the membrane coats for vesicular transport and endocytosis (COPI, COPII, clathrin) revealed that nucleoporin-encoding genes are distinctly co-expressed, suggesting a functional relationship (Supplementary Figs. 3 and 4).

I feel that additional follow up experiments would be needed to verify some of the key conclusions. For example, can the Y copy number (24 versus 32) be confirmed? Are the major biochemical interactions (that link individual subcomplexes within the NPC) conserved between algae/yeast/human or is the overall layout flexible (as the large divergence might suggest)?

The definitive proof for the number of Y-complexes in the human NPC was in situ structural analysis, similar to the work described in this manuscript for Chlamydomonas.

Quantitative mass spectrometry only measures relative protein abundance but not copies per NPC. Fluorophore counting might be used to address this issue, but both this method and mass spectrometry have a limited accuracy and are not commonly used to discriminate differences below a factor of two, as would be required in this case. Furthermore, to introduce a reasonable number of endogenously tagged fusion proteins into Chlamydomonas cells, which are not very straightforward to manipulate genetically, would be clearly beyond the scope of this manuscript.

Inter-subcomplex interactions would be expected to be more variable given that the overall architecture is not identical. Such contacts were best characterized for the inner ring of Chaetomium (Amlacher et al. Cell 2011; Fischer et al. Nature structural & molecular biology 2015; Lin et al. Science 2016; Teimer et al. Nature comm. 2017). The only known interactions that would be relevant for the point raised by the reviewer is the head-to-tail contact formed between Y-complexes in situ by Nup133 and Nup160. This contact has been observed in both yeast (Alber et al. Nature 2007; Seo et al. PNAS 2009) and humans (Von Appen et al. Nature 2015). We detected both Nups133 and Nup160 in the Chlamydomonas genome (see Supplementary Figure 1), and our fitted Chlamydomonas NPC structure is consistent with a head-to-tail contact between these two proteins (Supplementary Fig. 10).

As emphasized by our systematic fitting analysis, the three instances of the Y-complex observed in the Chlamydomonas are highly consistent with each other and with known

features of the Y-complex in other organisms. The appeal of in situ structural analysis is that it observes structural features within the native environment and thus overcomes many caveats underlying in vitro and biochemical analysis. We have revised the text carefully to remove any statements that might be over-interpretations. We hope that the reviewer agrees with us that our results are now presented appropriately. We believe the finding that inner ring architecture is highly similar yet dilated, whereas outer ring architecture varies considerably, is very exciting for the scientific field.

Reviewer #2 (Remarks to the Author):

Mosalaganti et al., describe the in situ structure of the nuclear pore complex from the green algae *Chlamydomonas reinhardtii* (CrNPC). The structure was obtained by aligning 78 subvolumes of nuclear pore complexes and then applying 8-fold symmetry. The resulting average shows clear structural features to 30 Å resolution allowing direct comparison with the human NPC (HsNPC). The authors discovered that while the overall architecture of the human and green algae NPCs were similar, several differences occurred at the subunit level. These evolutionary differences will greatly assist in deciphering how nuclear pores function.

The manuscript is well written with clear figures and explanations. The cryo-ET and subtomogram averaging is of high standard, the fitting of the atomic structures is clearly explained and the conclusions justified. The data presented is of high interest to the people in the NPC field and also researchers interested in the evolution of eukaryotes. Thus, I strongly recommend the publication of this manuscript.

We thank the reviewer for this positive assessment.

Minor comments:

line 243: remove 'is'

Line 262: remove 'be'

Supplementary figure 1: Perhaps choose a different color for Elys (e.g. green) as it took a while to find.

We have fixed these issues in the revised version.